# The History of the Brazilian Sardine (*Sardinella brasiliensis*) Between Two Fishery Collapses: An Ecosystem Modeling Approach to Study Its Life Cycle

**DOI:** 10.3390/biology14010013

**Published:** 2024-12-27

**Authors:** Rafael Schroeder, Angélica Petermann, Alberto Teodorico Correia

**Affiliations:** 1Laboratório de Estudos Marinhos Aplicados, Escola Politéccnica, Universidade do Vale do Itajaí (UNIVALI), Rua Uruguai 458, Itajaí 88302-901, Brazil; 2Centro Interdisciplinar de Investigação Marinha e Ambiental (CIIMAR), Terminal de Cruzeiros do Porto de Leixões, Avenida General Norton de Matos S/N, 4550-208 Matosinhos, Portugal; atcorreia@icbas.up.pt; 3Laboratório de Ecossistemas Aquáticos e Pesqueiros, Escola Politéccnica, Universidade do Vale do Itajaí (UNIVALI), R. Uruguai 458, Itajaí 88302-901, Brazil; angelicapetermann@gmail.com; 4Instituto de Ciências Biomédicas Abel Salazar (ICBAS), Universidade do Porto (UP), Rua de Jorge Viterbo Ferreira 228, 4050-313 Porto, Portugal

**Keywords:** pelagic fish, fisheries, population structure, ecopath, rational management

## Abstract

This study investigates the historical fluctuations in the Brazilian sardine (*Sardinella brasiliensis*) abundance and its impact on the purse seine fishery dynamics in southeast–south Brazil. Using an ECOPATH mass balance model, the trophic relationships and biomass importance between two fishery collapses, in 2000 and 2017, were analyzed. Following 2000, the fishery became multi-species. Mean trophic levels showed a decline until 2008, when more advanced vessels expanded the fishery grounds, increasing the trophic levels. However, by 2016, the fishery collapsed due to the high catches and natural mortality rates, a drop in primary production, and rising water temperatures, suggesting a detrimental cycle affecting the sardine stocks.

## 1. Introduction

Forage fish, such as sardines, bring multiple benefits, both to the production sector, through the exploitation of pelagic fish resources, and to the environment in which they are inserted [1,2,3]. In terms of the fish industry, pelagic species account for 30% of the global landings and are subsequently processed into food products and fish oil [4]. At the same time, these species are important components of marine food chains, as they create a link between lower trophic levels, consisting of planktonic organisms, and top trophic chain predators, such as large pelagic fish, birds, and marine mammals [5,6,7]. Understanding the trophic interactions of these species is critical to unravel their ecological role and significance in marine ecosystems [8,9,10]. Information on feeding habits and trophic positions was widely applied to a multi-species model, enhancing fish management and conservation measures [11,12,13]. Abundant medium-trophic-level ecosystem compartments, such as Brazilian sardine *Sardinella brasiliensis*, Atlantic thread herring *Opisthonema oglinum*, and chub mackerel *Scomber colias*, basically prey on low trophic level groups like phytoplankton and zooplankton. Ecosystem compartments of low- and medium-trophic-level groups are responsible for the main energy flows to higher trophic levels (e.g., squids, cutlass tail *Trichiurus lepturus*, king weakfish *Macrodon atricauda*, large pelagic fish, other finfish species and elasmobranchs), and also exert higher impact on the pelagic realm [14,15]. It means that it is important to understand the mechanisms that influence bottom–up control of the ecosystem structure in order to implement an ecosystem-based management approach [16].

The Brazilian sardine *Sardinella brasiliensis* is the main fishery resource off Brazilian waters, being caught typically between Cabo de São Tomé (Rio de Janeiro—22° S) and Cabo de Santa Marta Grande (Santa Catarina—29° S), with some unusual but expressive catches also in the south extreme of the Brazilian coastal waters [17,18,19,20]. The spawning of the species occurs in two main areas (23° S–25° S and 26° S–28° S) from early spring to early autumn at temperatures from 22 °C to 28 °C [17,21,22], which depends heavily on the water temperature and the seasonal advection of the South Atlantic Central Water (SACW) that modifies the structure of the environment, increasing the availability of food for the larvae and juveniles [22,23,24,25]. Brazilian sardine production reached a maximum of 228 thousand tons in 1973, declining progressively from this year on, reaching the first collapse in 1990, followed by a recovery of the stock until a new collapse in 2000 [17,23,24]. The collapse of the fishery was due to a failure in the recruitment caused by a very small spawning stock, not to mention the adverse environmental factors during the breeding season and the overfishing issues [18,23,26]. Possible effects for recruitment variation include: food selectivity, individual larval condition and related physical and biological processes, time follow-up of a cohort from spawn, and measure of vital parameters (survival and growth) along the spawning season and area, that could be enhanced by climate change and ocean conditions, which can promote periodic superabundance or absences of the Brazilian sardine [23,24,27]. From the 2000s onwards, *S. brasiliensis* stocks recovered reaching a stable level of around 100 thousand tons between 2012 and 2013, the highest production in the last 15 years. However, catches began to decline again in 2015, reaching the third collapse in 2017 [28].

Crises faced by the purse seine fleet, caused by the fall in yields of *S. brasiliensis*, have changed the behavior of this fishery [18,29,30]. From 1996 onwards, the purse seine fleet started to target other pelagic species, such as the Atlantic thread herring *Ophistonema oglinum*, the Atlantic chub mackerel *Scomber colias,* and the Atlantic bumper *Chloroscombrus chrysurus*, but also benthopelagic (*Selene setapinnis*) and demersal fish species (e.g., whitemouth croaker *Micropogonias furnieri* and catfish of the Ariidae family) [28,29,30]. In addition, some vessels started to use refrigerated brine to preserve fish onboard in 2008, instead of the conventional method of preservation in ice, increasing the autonomy of these vessels [28]. Despite the several historical modifications that have been conducted in the purse seine fleet, there are no estimates of such impacts over a long time series for the purse seine fishery in the *S. brasiliensis* ecosystem.

The ecosystem impacts of a fishery can be assessed through various indicators [1,6,31]. Modeling the energy flow in the exploited ecosystem allows assessing how the removal of component species can alter the trophic structure of this system [3,12,32]. In multi-species fisheries, the removal of large, long-lived fish with ‘slow’ life histories traits present at the top of marine food chains faster than their replacement capacity could cause them to decline more rapidly than species of smaller size, short-lived species with faster life cycles and lower trophic levels [33,34,35,36]. As a consequence, a gradual decline in the trophic level of exploited fish assemblages occurs, accompanied by a decline in the trophic level of the exploited ecosystem catch [34]. This phenomenon is known as “Fishing Down Marine Food Webs” [35]. Since it was proposed in 1998, this effect has been noticed in several studies conducted in several ecosystems [10,32,37]. This phenomenon became peculiarly known when the Biological Diversity Convention (2004) chose the trophic level of the catches as an index of diversity, defined as the marine trophic index [34], which describes the mean trophic level of the catches (MTL). Since then, various levels of MTL have been proposed and used to assess fisheries for low trophic level species of high abundance [34,37,38].

The MTL, however, can be masked by fleet technological improvements that can increase the stocking capacity of vessels or even improve the onboard fish preservation system (i.e., refrigerated brine fleet—VRB) [32,37,39]. Such modifications may result in an automatic gain in autonomy of the vessels that preserve fish in ice (ice fleet—VIC), allowing the fleet to expand geographically and bathymetrically into adjacent areas [40,41,42]. This expansion traditionally results in similar patterns, such as increasing catch and MTL with geographic expansion of the fishery [10,34,43]. To address this problem, a model was created that combines the properties of the MTL and an index that assesses the stability of the fishery relative to the trophic level (FiB) [34]. This index is designed to compensate for decreases or increases in the fishing area over time reflected in the trophic level of catches [11,42,44]. Therefore, for a given region, using FiB demonstrates the behavior of catches and the mean trophic level in a given year relative to the reference year to determine whether the change in the mean trophic level is compatible with the energy transfer efficiency (TE) in that region. This model, called the RMTL, allows us to describe changes in the MTL for distinct geographic regions in which expansion occurs [34]. Such expansions could be associated with the search for a distinct fishing area to obtain higher yields from a more abundant stock of *S. brasiliensis* [19,20,28].

The total biomass of the system can be modeled by the intensity of trophic relationships between the species present and how much the biomass of each contributes to the total biomass of the system [3,8,32]. Thus, mapping the energy flux from the diet of the species present would help to determine the trophic position of the component species and their influence on the two sides of the coin, both as predator (impacting), and as prey (impacted) [7,10,13]. Some species may also exert an enormous influence on ecosystem dynamics and the abundance of other species [12,45]. In this context, the hereby study evaluated the diet composition of the main species of the Brazilian sardine fishery in southeast–south Brazil based on information available in the literature to: (i) reconstruct the trophic level of the catches and the spatial and temporal variation; (ii) to model the direct and indirect impact of the purse seine fishery on other species that are interconnected in the trophic web; and (iii) to evaluate the presence of species of high impact in the ecosystems exploited by the Brazilian sardine purse seine fishery and its influence on the trophic web generating subsidies for a better regional fisheries management.

## 2. Materials and Methods

### 2.1. Study Area, Biological, and Fishing Data

The study area is located on the southeast and south Brazilian coast, from Rio de Janeiro (22° S) to Rio GrxB0; S) to Rio Grande do Sul (34° S), between depths of 100 and 200 m (Figure 1).

The total area can be subdivided into three macro-regions according to the latitudinal distribution of the oceanographic characteristics [47,48,49,50]: (i) a northern area, from Cabo de São Tomé (22.04° S) to Cabo Frio (22.87° S) in the state of Rio de Janeiro; (ii) a central area, located between Cabo Frio (22.87° S) and Cabo de Santa Marta Grande (28.63° S), between the states of Rio de Janeiro and Santa Catarina; and (iii) a southern area, which includes the southern Subtropical Shelf (between 28.63° S and 34.00° S) distributed along the states of Santa Catarina and Rio Grande do Sul (Figure 1). In the study area, the Brazil Current (BC) transports southwards the warm and salty oligotrophic tropical water (TW) along the entire slope from the upper layer of the water column down to 200 m depth, contrasting with the SACW present in deeper waters (>200 m depth) [48,49,50]. The cold and nutrient-rich SACW is upwelled in the northern area due to wind and topographic effects [49]. The continental shelf is characterized by the presence of the warmest and salty shelf water (SW), resulting from the mixture of the continental runoff (CR) with SW, being also influenced by the BC [50]. In the central area, the SW occupies its northern portion, while the subtropical shelf water (STSW) is present in its southern portion. STSW originates from the mixture of Plata plume water (PPW) and tropical water (TW), which spreads northward throughout the neritic region from the south to the central area [48,49]. In the northern portion of the central area, SACW contributes to a high concentration of nitrates, namely in two specific sites, near Cabo Frio and Cabo de Santa Marta Grande [48,50]. In the southern area, including the southern portion of the central area, the spread to the north of relatively cold and fresh PPW transport is more intense during the austral winter [49], when the sub-Antarctic shelf water (SASW) is also transported by the Patagonian Current (PC), which forms a wedge of cold water between PPW and STSW [48,49,50]. PPW and the austral waters of SASW contribute to an important input of a high concentration of silicates and phosphates associated with the continental runoff [50].

Biological and fishing information was obtained during landings from the commercial catches of 100 purse seine vessels with onboard fish preservation in ice and 10 purse seine vessels with preservation in refrigerated brine that operated in southeast–south Brazil between 22.60° S–30.25° S and at 7–110 m depth from 2000 to 2019 as part of the daily sampling program of the industrial fishery, and from scientific projects intended to assess the biomass and population structure of the Brazilian sardine [19,20,28,46,51,52].

Fishery data were collected during the landings within the daily program of industrial fishing monitoring in Santa Catarina, Brazil. During landing, information on the catches of *S. brasiliensis* and other species retained, the date (year and month of the capture and days at sea), and the geographical position of the catch (latitude, longitude, and depth), were obtained from interviews with vessel skippers. After collection, these data were verified and validated by trained professionals and incorporated into a database.

Data information collection was conducted from 2000 to 2019 from purse seine vessels with onboard fish preservation in ice (VIC) that exploited *S. brasiliensis* along with other species. Information from refrigerated brine vessels (VRBs) started in 2007, coinciding with the very beginning operation of highly technological purse seine vessels in southeast–south Brazil [20].

### 2.2. Variability of the Trophic Level of the Catches on a Spatial and Temporal Scale

#### 2.2.1. Mean Trophic Level of the Catches

The trophic level of catches from the *S. brasiliensis* fishery was determined [34,53,54], in order to verify the occurrence of the replacement of large predatory fish at the top of the trophic chain, by lower trophic level species [35]. This analysis considered the species composition landed between 2000 and 2019. For each species i, the trophic level (TLi) is defined by the following:(1)$${TL}_{i}=1+\sum_{j=1}^{n}{CA}_{ij}{TL}_{j}$$
where CA_ij_ represents the food composition (diet), j is the prey, and TL_j_ is its trophic level. Estimates of TL_i_ were searched in the literature and complemented by available information (Table 1). Considering Y_ik_ as the reported catches in year k of all species i with trophic level NT_i_, the mean trophic level (MTL) of the catches in year k was then estimated from the following:(2)$${MTL}_{k}=\frac{\sum_{i=1}^{m}Y_{ik}{TL}_{i}}{\sum_{i=1}^{m}Y_{ik}}$$
where Y_ik_ are the landings of species i in year k, and m is the number of species or groups of species caught in year k [54,55]. Given the efficiency of energy transfer between trophic levels TE, one index that assesses the stability of the fishery relative to the trophic level (FiB) was determined [11]. FiB was calculated to verify whether changes in the mean trophic level were balanced by changes in catch over this period by evaluating the expression of [43]:(3)$${FiB}_{k}={log}_{10}\left[Y_{k}{\left(\frac{1}{TE}\right)}^{{TL}_{k}}\right]-{log}_{10}\left[Y_{0}{\left(\frac{1}{TE}\right)}^{{TL}_{0}}\right]$$
where *k* refers to year zero, corresponding to the reference year (2000), Y = capture, MTL = mean trophic level of capture, and TE = efficiency in energy transfer between trophic levels. Y_0_ and MTL_0_ are the catch and the mean trophic level in the reference year, respectively. For TE, a value of 0.1 was used, a value derived from models applied to 48 ecosystems [14]. The fishery is said to be in equilibrium when FiB remains equal to zero, i.e., the catch falls proportionally on all trophic levels present, according to their availability in the environment. On the other hand, positive FiB values indicate the geographic and/or bathymetric expansion of the fishery [34].

#### 2.2.2. Mean Trophic Level of the Catches

The numerical representation of the expansion of the fishing areas was described by the variation of FiB. It is important to consider that FiB depends on the accuracy of MTL_0_ in reflecting the average trophic level of species available in the initial fishing area [34]. It is possible that at the beginning of the fishery, the fleet did not exploit the full spectrum of available species, due to the collapse of *S. brasiliensis* this year, which allows the assumption that the range of the purse seine fleet in southeast and south Brazil increased its distribution. To evaluate this effect, an index called region-based marine trophic index (RMTL) permitted to recalculate the MTLs [Equations (5)–(7)] for the years in which the fishery increased this radius, as proposed by [34]. According to the authors, this estimate is based on two assumptions: (1) that stocks in the initial area continue to be fished, and (2) that the fishery in the initial region remains in stability with respect to trophic level. To estimate the catches and associated MTLs for the regions of fishery expansion in years (k), the initial year is taken as a reference, i.e., the initial value of FiB equals zero and using the resulting MTL (MTL_k_):(4)$${\hat{Y}}_{k}^{1}=Y_{n_{1}}*{\left(\frac{1}{TE}\right)}^{{MTL}_{n_{1}}-{MTL}_{k}}\;for\;k>n_{1}\;$$

The differences between the reported catches Y_k_ in year k compared to the previous year represent the catches in the expanded fishing area:(5)$${\hat{Y}}_{k}^{r}=Y_{k}-\left({\hat{Y}}_{k}^{1}+\ldots +{\hat{Y}}_{k}^{r-1}\right)\;for\;n_{r-1}<k\leq n_{r}$$

Thus, the mean trophic level of the initial region is calculated by setting FiB equal to zero and using the reported catches (Y_k_):(6)$${\hat{MTL}}_{k}^{1}=\left[{MTL}_{n_{1}}-\frac{{log}_{10_{\frac{\left(Y_{k}\right)}{Y_{n_{1}}}}}}{{log}_{10_{\left(\frac{1}{TE}\right)}}}\right]$$

Using the expression above, the MTL of the expanded region is calculated as follows:$${MTI}_{k}*Y_{k}=\sum_{i}Y_{ik}*{NT}_{i}={\hat{MTI}}_{k}^{1}*{\hat{Y}}_{k}^{1}+{\hat{MTI}}_{k}^{2}*{\hat{Y}}_{k}^{2}$$

From the above equality, the estimated MTL could be calculated for the expanded region:(7)$${\hat{MTL}}_{k}^{2}=\frac{{MTL}_{k}*Y_{k}-{\hat{MTL}}_{k}^{1}*{\hat{Y}}_{k}^{1}}{{\hat{Y}}_{k}^{2}}$$

The variation of FiB was confronted with the geographic representation of the fleet’s operation area. The fishing areas obtained from interviews during landings were geographically represented in quadrants of 30’ visited on each fishing trip and expressed for the years monitored, and the variation in the number of quadrants visited. Considering that fishing in a more distant area would imply a longer travel time and larger and more powerful vessels, other indexes were constructed considering information on vessel length, gross tonnage, and engine power obtained from the General Fisheries Register since 1938. In parallel, indices of (i) days at sea, (ii) fishing days, and (iii) number of fishing operations are in relation to the reference year.

### 2.3. Modeling Trophic Relationships

The trophic structure of the environment exploited by the purse seine fleet was described based on the commercial species that landed in Santa Catarina [28] and on information available in the literature (Table 1). The food web was modeled using Ecopath with Ecosim (EwE), refs. [2,11,13] based on five previous determinations applied in the study area [7,14,15,52,57]. The trophic relationships were described by fitting simultaneous linear equations for each group *i* in the system:(8)$$P_{i}-B_{i}*M_{2i}-P_{i}\;\left(1-{EE}_{i}\right)-{EX}_{i}=0$$
where P_i_ = production of i (g m^−2^ yr^−1^); B_i_ = biomass of i (g m^−2^); M_2i_ = predation mortality of the i (yr^−1^); EE_i_ = ecotrophic efficiency of (fraction of 1); 1 − EE_i_ = other sources of mortality (yr^−1^); EX_i_ = exportation of i (g m^−2^ yr^−1^). The total production of group i is controlled by predation by other groups (B_i_M_2i_), non-predation-related losses [P_i_ ∗ (1 − EE_i_)], and for losses to other systems (EX_i_). Production was estimated from the production rate per biomass (PB_i_) and annual mean biomass (B_i_), which can be expressed as P_i_ = Bi ∗ PB_i_. Predation mortality depends on predator activity and can be expressed as the sum of the consumption of all predators (j) preying on a group (i):(9)$$\left(B_{i}*M2i\right)=\Sigma jBj*\frac{Q}{Bj}*DCji$$
where QB_i_ = biomass consumption rate of predator i (yr^−1^) and DC_ij_ = fraction of prey i in the diet of predator j. Thus, Equation (1) can be rewritten as follows:(10)$$\left(Bi*\frac{P}{Bi}*EEi-\Sigma jBj*\frac{Q}{Bj}*DCji\right)=0$$

#### 2.3.1. Model Input Parameters

The initial parameters B, BP, QB, and EE were inserted for each group, while the other parameters were calculated using the software (Table 2). The trophic compartments of the system were composed of the main species caught by the purse seine fleet based in Santa Catarina between the years 2000 and 2016 and the respective food items for which information was available (Table 1). For each recorded species, a detailed dietary search was carried out containing the percentage share of each food item. Because the available diet studies have more than one description per species, the available surveys were compiled into a single determination. For each species’ diet composition, the food items were normalized [56] and a single diet composition was created (Appendix A). The aggregated dataset for each species was adjusted to a maximum likelihood estimation method that produces probability density functions and associated error ranges for each predator and prey interaction, in which each function indicated the fraction of the prey’s contribution to the predator’s diet. Species’ diet was validated by comparing them to associated error ranges [56].

Biomass was calculated in tons per square kilometer (t/km^2^). For the species selected, the annual biomass Bs,y of species s caught by the purse seine fleet in southeast–south Brazil in year y was determined by summing the catch in all quadrants visited, where Quad is the number of quadrants in the study area [58]:(11)$$B_{s,y}=\sum_{k=1}^{Quad}B_{s,k,y}$$

Density estimates (t/km^2^) were obtained by dividing the biomass by the total area of the quadrants visited. Biomass production, however, varies between different species [11]. Fish growth strategies were described by the von Bertalanffy growth curve, which depends basically on two parameters: the growth rate coefficient (K), and the asymptotic length (L_∞_), defined by the following:(12)$$L_{t}=L_{\infty }*(1-e^{-k\left(t-t_{0}\right)})$$
where *L_t_*—fish length (cm); L_∞_—asymptotic length (cm); K—growth rate coefficient (year^−1^); T_0_—nominal age (year) when fish size is considered to be zero. Estimates of these parameters assist in fisheries management by helping to calculate other population attributes such as natural mortality (M), production per biomass (PB), and consumption rates per biomass (QB). Natural mortality was calculated by the empirical relationship of [59]:(13)$$LogM=0.654\;LogK-0.279LogL_{\infty }+0.463LotT$$
where M—natural mortality (year^−1^); L_∞_—asymptotic length (cm); K—growth rate coefficient (year^−1^); T—mean annual water temperature of the study system. The values of K and L_∞_ were acquired from growth studies of the species used in the model. Fish production is the total amount of tissue produced by a population during a given period of time, even though not all individuals survive to the end of that period demonstrated that under certain equilibrium conditions, P/B (production per biomass) is equivalent to total mortality (Z), and this is defined as Z = M + F, where F is the fishing mortality per year [11].

Determination of fishing mortality requires the quantification of the biomass of organisms of different age classes exploited by fisheries during a given time period [60]. In this regard, estimates of Z were obtained from longevity (Tm) observations in the model developed by Hoenig in 1983 [61]. This model is described by the linear equation based on observations of Z and Tm:(14)lnZ=a+b∗lnTm
where a = 1.46 and b = −1.01 for fish (based on 84 stocks of 53 species) and a = 1.23 and b = −0.832 for mollusks (based on 28 stocks of 13 species). The QB calculation followed the empirical method of [62]:(15)$$LogQB=7.964-0.204\;LogW_{\infty }-1.965\;T^{\prime }+0.083Ar+0.532H+0.398D$$
where Q/B, annual consumption/biomass ratio; W∞, asymptotic weight (grams wet weight); T′, inverse of mean water temperature = [1000/(23.4 °C + 273.15)]; Ar—index of caudal fin shape: Ar = h^2^/S; S, fin surface area (mm^2^); H and D—index of food type: H = 1 for herbivores; D = 1 for detritivores and iliophages while for carnivores H = D = 0 [11].

The primary production and detritus flow from trophic level 1, which are required to sustain fisheries (PPR, gCm^−2^ year^−1^), permits the evaluation and comparison of the fishing activities across ecosystems. The PPR increases with fishing intensity, and is obtained by back-calculating the flows in primary production and detritus equivalents for all pathways of the species caught down to the primary producers and detritus [6,11,55] using the following formula:(16)$$PPR=\sum_{paths}\left[\frac{Y_{i}}{P_{i}}\;\times \;\prod_{j,i}\frac{Q_{j}}{P_{j}\;\times \;{EE}_{j}}\;\times \;{DC}_{j,i}\right]$$
where Y_i_ is the catch of a given group i, P represents the production of predator J, Q the consumption of predator j, DC the diet composition of each predator j/prey i interaction in each path, and EE is the ecotrophic efficiency, or the proportion of the production that it is used within the system due to consumption or is exported from the system (e.g., in terms of catches).

EwE estimates a series of overall ecosystem attributes such as the biomass in habitat area (t/km²), consumption biomass (year^−1^), ecotrophic efficiency, production consumption (year^−1^), biomass accumulation (t/km^2^), biomass accumulation rate (year^−1^), flow to detritus (t/km²/year), net efficiency, omnivory index, fishing mortality/total mortality, and proportion natural mortality. These variables were used to investigate possible ecosystem differences between the original and the expanded fishing area.

#### 2.3.2. Impact on Trophic Relations

The effect that changing the biomass of one group has on the biomass of the others was estimated using a method called the mixed impact trophic matrix [5,7,15]. The set of elements in this matrix (i,j) represents the interaction between impacted group i by impacting group j, as described by the following:(17)$${MTL}_{i,j}={DC}_{i,j}-{FC}_{j,i}$$
where DC_i,j_ expresses how much species j contributes to the diet of i and FC_j,i_ gives the proportion of the predation of j that is attributed to predator i. In a balanced trophic model, the mixed impact on the trophic web is estimated for each pair (i,j), regardless of whether they interact directly or not. Because the species present are interconnected, the effect was scaled for impacting species (row) on impacted species (columns). These values are expressed in percentages, where negative values indicate the negative impacts of predators on prey (top–down control). On the other hand, positive effects indicate the prevalence of effects of the prey on the predator (bottom–up control) [5,7,15].

### 2.4. Identification of Keystone Species

Keystone species were identified based on the classification tree-based method among the functional groups of each modeled ecosystem [12]. This technique used log-transformed values of the trophic impact (in squared values) and biomass (Bi) of each group i. Species (or model compartments) were further classified into four categories: keystone (corresponding to groups with high impact and low biomass), low-impact–low-biomass, low-impact–high-biomass, high-impact–high-biomass, and intermediate (corresponding to groups belonging to none of the previous categories).

### 2.5. Statistical Analyses

The variability of MTL between 2000 and 2016 was evaluated using a generalized additive model for location scale and shape (GAMLSS), following previous studies for *S. brasiliensis* [12,63]. MTL was confronted in a temporal perspective to FiB, fishing area [original area (2000–2007) and the expanded area (2008–2016)], fleet (ice vessel fleet—VIC, and refrigerated brine fleet—VRB), year, total catch, and variables estimated by EwE for the original and the expanded area: PPR and flow to detritus.
MTL=FiB+Fishing area+Fleet+Year+Total catch+PPR+Flow to Detritus

Posteriorly, output variables from the mass balance model, such as biomass in habitat area, biomass, biomass (production, consumption, and accumulation), ecotrophic efficiency, flow to detritus, omnivory index, fishing mortality per total mortality, and proportion natural mortality and trophic impact over keystone species were compared between two periods: (i) 2000–2007 and (ii) 2008–2016, using a Kruskal–Wallis test. These analyses were followed by a permutational multivariate analysis of variance (PERMANOVA) to evidence differences between those periods [64]. For PERMANOVA analyses, the dissimilarity matrices were based on altGower distance, and the p-values were generated using 9999 permutations using the vegan package.

## 3. Results

### 3.1. Variability of the Trophic Level of the Catches on a Spatial and Temporal Scale

A total of 97 species landed in Santa Catarina between 2000 and 2019 were used to calculate the trophic level of the catches. The description of the MTL of the catches landed by the purse seine fleet in southeast–south Brazil (Figure 2A), showed great oscillation between the years 2000 and 2019 (Figure 2B). The index that describes the balance of the fishery in relation to trophic level (FiB), on the other hand, showed positive values, with an upward trend until the year 2012, decreasing from then until 2019 (Figure 2C). The RMTLs showed similar variation in the initial area (2000–2007), and in the expanded area (2008–2016), as it showed no significant difference in the test of homogeneity of slopes (df = 2, F = 1.929, *p* = 0.174). The GAMLSS model, on the other hand, showed significant differences for the MTC for the refrigerated brine fleet (Figure 2D), and all other variables, namely FiB (t = −2.854, *p* = 0.021), fishing area (t = −5.243, *p* = 0.001), total catch (t = 2.601, *p* = 0.031), and PPR (t = 2.852, *p* = 0.021), except for flow to detritus (t = 0.141, *p* = 0.891).

These positive values of FiB indicate an expansion of the fishing area, relative to the reference year (2000). Such an increase becomes clearer through the reconstruction of the geographical position of the catch and the number of quadrants fished (Figure 3). Until 2007, the sardine fishery was concentrated between the northern region of the State of São Paulo and the central region of Santa Catarina. From 2007, the vessels that landed in Santa Catarina expanded the area of operation between the northern limit of Rio de Janeiro (21° S) and the extreme south of Brazil in the state of Rio Grande do Sul (33° S) (Figure 3).

Analysis of the physical characteristics of the vessels licensed to catch *S. brasiliensis* showed a progressive increase in length, gross tonnage, and engine power from 1938, but this trend has been more pronounced since 2008 (Figure 4A–C). For vessels with fish preservation in ice, the values for days at sea, fishing days, and number of hauls showed a progressive decrease from the reference year, resuming values similar to the initial ones in 2015 and 2016 (Figure 4D–F). Refrigerated brine vessels showed a decline in the number of hauls and an increase in days at sea and fishing days (Figure 4D–F).

The values of *S. brasiliensis* catches by fishing area showed a peak at 26 °C between 2000 and 2007, which was maintained in the following years (2008–2016, Figure 5). In this second period, a subtle increase in catches by the ice fleet was detected at 23^o^ S. The refrigerated brine fleet, on the other hand, obtained their catches at 23° S, with a less abundant peak at 26° S between 2008 and 2016 (Figure 5).

### 3.2. Modeling Trophic Relationships

In terms of diet, *S. brasiliensis* basically consumes zooplankton (70.3%) and phytoplankton (29.7%) (Table 1). Information on its participation as a food item, including percentage participation, was found for the skipjack tuna (5.6%), the dolphinfish (5.2%), and the slender inshore squid (0.03%). The reconstructed trophic relationships for the other species included in the ecosystem model were jointly presented (Table 1, Figure 6).

The sum of the effect of direct and indirect trophic interactions obtained from the mixed-impact trophic matrix was negative indicating the prevalence of top–down control effects (Figure 7). The Brazilian sardine was the species that showed the highest number of interactions among the trophic compartments considered (Figure 7). The main pelagic species also showed a large number of interconnections, such as the Atlantic thread herring, the Atlantic bumper, and the Atlantic chub mackerel. Among the trophic compartments that constitute the food items of the species entered in the model, the ones that showed the highest number of interactions were benthic-feeding fish and pelagic fish (Figure 7).

Considering the Brazilian sardine as an impacted species, negative impacts are observed on the Atlantic thread herring, pelagic fish, Atlantic moonfish, Atlantic bumper, Atlantic chub mackerel, and detritus. These species are impacted indirectly, either by being associated in the same catch area as the Brazilian sardine or by top–down control effects on prey. The dolphinfish, lebranche mullet, and planktonic organisms, such as phyto- and zooplankton, are positively impacted by bottom–up processes (Figure 7). As an impacting species, the Brazilian sardine exerts negative impacts on all landed species except the Largehead hairtail. Positive bottom–up control effects occurred on all compartments represented by food items with the highest values observed on organisms associated with the seabed such as shrimp, mollusks, polychaetas, cnidarians, crabs, and benthic-feeding fish (Figure 7).

### 3.3. Identification of Keystone Species

In the original area, represented by the classification tree (2000–2007), five species fell into the category of hi-impact–hi-biomass: yellowfin amberjack (OL), Atlantic bumper (PA), whitemouth croaker (CO), Brazilian sardine (SV), and phytoplankton (FIT); four were characterized as low-impact–low-biomass [crabs (CAR), cnidarians (CNI), bacterioplankton (BAP), and algae (ALG)]; and three of them as keystone species [dolphinfish (DO), pelagic-feeding fish (PAP), and shrimps (CAM)] (Figure 8). In the expanded area, represented by the classification tree (2008–2016), five species were placed in the category of hi-impact–hi-biomass: Atlantic thread herring (SL), rough scad (XI), Atlantic chub mackerel (CA), and Brazilian sardine (SV); four fell into low-impact–low-biomass [crabs (CAR), cnidarians (CNI), bacterioplankton (BAP), and algae (ALG)]; and two of them were characterized as keystone species [benthic-feeding fish (PAB) and pelagic fish (PEP)] (Figure 8). The other species did not match any of these categories being classified as intermediate. Significant differences were observed between the original and expanded areas for the trophic impact over species of hi impact and hi biomass. Trophic impact for species of hi impact and hi biomass was higher in the expanded area (Kruskal–Wallis, chi-squared = 5.7709, df = 1, *p*-value = 0.01629, Figure 8). No significant trophic impact was detected for the other categories.

### 3.4. Influences on the Trophic Web

The original and expanded areas also showed univariate significant differences for biomass, flow to detritus, fishing mortality per total mortality, and proportion of natural mortality (Figure 9). These areas also showed significant differences on a multivariate scale (PERMANOVA, *pseudo-F* = 3.7386, *p* = 0.007). On a temporal scale, PPR values were minimal during the reference year (2000). These values increased progressively until 2008, maintaining high values until 2012. Afterwards, a sudden drop was observed (Figure 9). In the year 2012, the highest sardine landings were recorded, showing a decreasing trend thereafter until the collapse of 2017.

## 4. Discussion

The reconstruction of the MTL obtained from the commercial catches of *S. brasiliensis* ecosystems in the purse seine fleet operating during the last two decades off southeast–south Brazil showed a clear up-and-down variation pattern. The interpretation of such patterns may result from combined effects of distinct fishery grounds within the entire species distribution area with the following: (i) the existence of distinct biological population units associated with particular spatial/temporal distribution fish patterns, namely the south population unit distributed in the central and southern part of the study area (24° S–32° S), and the north population unit distributed in the northern part of species distribution area (22° S–23° S); (ii) the spatial/temporal fishing strategies involving VRB may be associated to distinct biological patterns of the north stock, while the south population-unit is mostly explored by VIC; and (iii) the effects of the environmental pressures, as such the spatial and temporal variability in the PPR.

MTL is often used as an indicator of ecosystem health [35,65,66]. The higher value of trophic level observed in the reference year (2000) probably occurred due to the catch of other species induced by the low abundance of the target species in a post-collapse period. With the gradual recovery of *S. brasiliensis* stocks, the mean trophic level of the catches progressively decreased until 2007, approaching the trophic level of *S. brasiliensis*. The increase in MTL values from 2007 onwards, which coincided with the increase in positive FiB values, may represent an increase in the primary productivity of the environment (bottom–up control effect), but is more commonly described by the expansion of the exploited ecosystem when associated with the increase in MTL [10,34,44].

The decline in the MTL usually occurs as a result of the selective removal of a high percentage of large species, whereas a stable MTL demonstrates that all trophic levels are being sampled equally [10,65,66]. Preferential removal of these species may result in a disruption in the structure and function of exploited ecosystems (i.e., ecosystem overfishing) [28,36,67]. Ecosystem degradation can be best measured using environmental indicators, among them the primary production required to sustain the exploited biomass (PPR), which depends on accurate estimates of productivity and energetic requirements of the species caught, unless the discards are minimum, as observed in the *S. brasiliensis* fishery [1,5,20,68]. The determination of these parameters demonstrates the difficulty in detecting specific impacts of fishing, because in areas of higher productivity and intense fishing exploitation, the decline in the MTL may be associated with the heterogeneous effect of productivity along different trophic levels [1,11]. However, the decline in the FiB index observed in some areas of the Mediterranean Sea concomitant with a steady decline in MTL has been associated with catch declines induced by unsustainable levels of fishing [10].

A geographical and bathymetric expansion of the fishing pressure on all trophic levels of the marine food web in the Mediterranean Sea, coincidental with the introduction of new technologies in both vessels and gears, increased the impact on exploited stocks and habitats in the second half of the 20th century [10,32,69]. This expansion has been reflected in the decline in the landed volumes and mean trophic level of catches over the last three decades. On this occasion, the MTL suggested the prevalence of unsustainable fishing exploitation regimes or poor conditions of the state of fish stocks, typically occurring in the most intensively exploited regions [10,70,71]. However, the characteristics of the pelagic ecosystems of southeast–south Brazil seem to be associated with fishing through the food web effect, which is characterized by the introduction of technological artifacts in the fisheries, which increase the catchability of the target species, and consequently leads to a reduction in the MTC approaching to the target’s species [32,65,68]. The magnitude of the new technological artifacts is more sensitive during moments of low stock productivity and abundance levels of small pelagic fish. In such cases, simulations of fishing suspension or restricted to low levels (i.e., one-half of the average catch) showed positive results in stock biomass recovery and the reduction in the number of collapsed stocks [65].

Regardless of the effect, this greater coverage of the fishing area from the same port shows a change in the dynamics of the fleet, allowing it to operate in more distant fishing areas and return to unload in the port of origin [28]. Even though the areas expanded from 2008 are not unexploited areas [51], the MTL calculated separately by area (RMTL), revealed that the trophic level exploited until 2007 and in the more distant fishing areas starting from 2007 showed a significant difference. The adjusted RMTL values obtained by different purse seine fleets and fishing areas evolved towards the trophic level of *S. brasiliensis* (target species). This difference in the trophic level in the expanded area, resulting from the operations of the refrigerated brine fleet has evidenced a distinct species composition exploited from 2008 onwards can be distinguished by the significant high trophic impact over species characterized as hi-impact-hi-biomass. At the same time, this expansion may result in a possible deleterious effect on the exploited ecosystems [1,5,31], namely when fisheries move forward to other areas or deeper regions [39,40,42]. This expansion to more distant fishing areas despite possibly favoring obtaining better yields was mostly associated with catches of Brazilian sardines in deeper fishing grounds [19,20], previously unassessed by the ice fleet. The increased fishing effort in the northern region (expanded) may have contributed to the collapse of *S. brasiliensis* in the year 2017 [65], since this stock is basically supplied by self-recruitment, but also makes a high contribution to all sardine fishing areas (see [47] for details). As the decline of the FiB index in southeast–south Brazil, accompanied by a concomitant declining MTL trend, can be attributed to the decline in catch and indicates unsustainable fisheries, these two indexes could be used as continuous indicators to monitor and manage fishing pressure in the expanded fishing areas.

The upgrading from part of the purse seine fleet to incorporate the refrigerated brine system led this fleet to obtain greater autonomy and fishing power; as a consequence, it may represent a complementary strategy of the canning industry to obtain a more abundant source for the industry [19,20]. This change may also be associated with the prohibition of the capture of demersal fishes such as the whitemouth croaker *M. furnieri*, the argentine croaker *Umbrina canosai*, the king weakfish *Macrodon atricauda* and the stripped weakfish (*Cynoscion guatucupa*) from 2007 onwards [72]. Nevertheless, even with the ban on catching several demersal fish, benthic-feeding fish continue to be indirectly impacted via interconnected trophic relationships [12,46,48] in the ecosystem exploited by the Brazilian sardine fishery.

In terms of the Brazilian sardine biomass, the two peaks of abundance found in the hereby study (one in the original fishing area and the other in the expanded one) have demonstrated the existence of the different stocks previously defined [21,73,74], and posteriorly confirmed through the analyses of the body morphometrics, otolith shape and microchemistry [20,28,52]. A long-term analysis using simple biological and reproductive data of *S. brasiliensis* has validated previous population structure studies [19]. This latter study has also evidenced the influence of the seawater temperature in the northern area of species distribution. This increase in temperature was associated with a process of tropicalization of the South Atlantic Ocean with the intensification of the Brazil Current [75,76,77]. The effect of tropicalization may have left a chemical signature on the otoliths of *S. brasiliensis* juveniles described by the high quantity of Mg/Ca [47] and on the size of first maturity, since the stages of maturation most sensitive to reproduction were associated with colder waters [19]. In addition, this anomalous warming of the southwest Atlantic waters was also negatively correlated with the recruitment of *S. brasiliensis* [19], as suggested in previous studies [18,23,26]. The Brazilian sardine is a summer spawning species that lays eggs in synchrony to the upwell of enriched deep waters of the SACW [17,20,21]. The SACW modifies the structure of the environment, increasing the availability of food for the larvae and juveniles [28], therefore, processes that disturb Ekman transport, as such sea surface temperature and wind stress, can directly influence the Brazilian sardine biomass and natural mortality, that is directly linked to the Brazilian sardine recruitment [18,23,25].

In recent years, one of the world’s largest marine warming hotspots extending from Cabo Frio (Brazil, 22° S) to Tierra del Fuego (Argentina, 55° S) was detected, as its water temperature has increased above the global average [78,79]. Satellite-derived sea surface temperatures have shown positive anomalies of 0.5 °C between 2000 and 2016 in this region [75,78]. Such anomalies have induced a poleward displacement of the wind patterns over the South Atlantic which led to a southward expansion of the warm waters of the Brazil Current (BC) over the past decades, and a subsequent warming along its path [75,80,81]. Long-term observations on estuaries and sandy beaches at Sepetiba Bay (~23° S), which serve as rearing grounds for commercially important fishes such as *S. brasiliensis*, suggest that boundaries of fish fauna distribution may have displaced poleward [26], presumably in response to ocean warming [76]. Moreover, changes in the presence and relative abundance of *S. brasiliensis* over four decades (1980–2010), suggest that the region is facing a “tropicalization” of the marine community and *S. brasiliensis* is expanding its habitat southwards [19,26,75]. The plausibility of the “tropicalization” process in the region agrees with the poleward expansion of tropical fish [75,77]. These thermal anomalies can modify the primary production required to sustain the exploited biomass in southeast and south Brazil, which can influence the feeding conditions of the environment, causing excessive natural mortality to juveniles of *S. brasiliensis* [22,23,24]. It is possible that the drop in PPR observed in the hereby study is associated with the tropicalization of the South Atlantic, since it began in synchrony with the increase in the average temperature of the catches described in other studies for the same study area [19,26,75,76,77,78,79]. The effects of a poleward shift in the trailing range edge of warm-water affinity species as such *Sardinella brasiliensis*, *Anchoa lyolepis*, *Anchoa tricolor*, and *Harengula clupeol*) responded to ocean warming with faster population growth rates, whereas cold-water affinity species (e.g., *Anchoa marinii*, *Anchoviella brevirostris*, *Anchoviella lepidentostole*, and *Lycengraulis grossidens*) will disappear or drastically decreased their abundance [75].

## 5. Conclusions

The hereby study reinforces the existence of two stocks of the Brazilian sardine, which is still managed as a single stock [19,20,28,47]. Both stocks are suffering from different fishing and environmental pressures. During the collapse of the Brazilian sardine fishery in 2000, the south stock was most severely affected, while the north stock showed a healthier condition in the early 2000s [19]. Thus, there seems to be a modification in the environmental conditions in southeast–south Brazil that reminds a “see-saw” movement, which may depend on various factors, but mainly on long-term oscillation effects [18,19,22,23,24]. Such long-term oscillations combined with escalating fishing efforts in the northern (expanded) fishing area may have contributed to the collapse of *S. brasiliensis*. The continuous evaluation of the long-term temporal and spatial stability of the population structure of the Brazilian sardine stocks subject to exploitation is of utmost importance [47], which may be amplifying its distribution to south Brazil. Future studies should focus on habitat suitability models of *S. brasiliensis* distribution in relation to the climate-changing conditions of the southwest Atlantic Ocean serving as subside for the implementation of an action plan that includes adaptive management frameworks for fisheries. A recently developed study sought not only to generate up-to-date data and information on the fishing resources and demersal fisheries of the Brazilian Meridional Margin, but above all to develop a new fisheries management ecosystem-based model more suited to the complexities of a multi-species and multi-fleet fishery, developed in a region of high biological diversity [82]. This plan suggested the implementation of spatial management units based on positive examples used elsewhere on the planet. A second edition of this project will include stock assessments for *S. brasiliensis*, and the implementation of a spatial management unit for pelagic fisheries, as was recently proposed for the demersal fisheries. The findings obtained in the hereby study would serve as a baseline for the definition of the spatial management units that could be used to establish regional management measures and, as such catch quotas per each stock identified. Periodic stock-specific assessments should be conducted altogether with the indexes applied (FiB and MTL) to continuously evaluate the healthiness of the most important pelagic resource of Brazilian waters.

## Figures and Tables

**Figure 1 biology-14-00013-f001:**
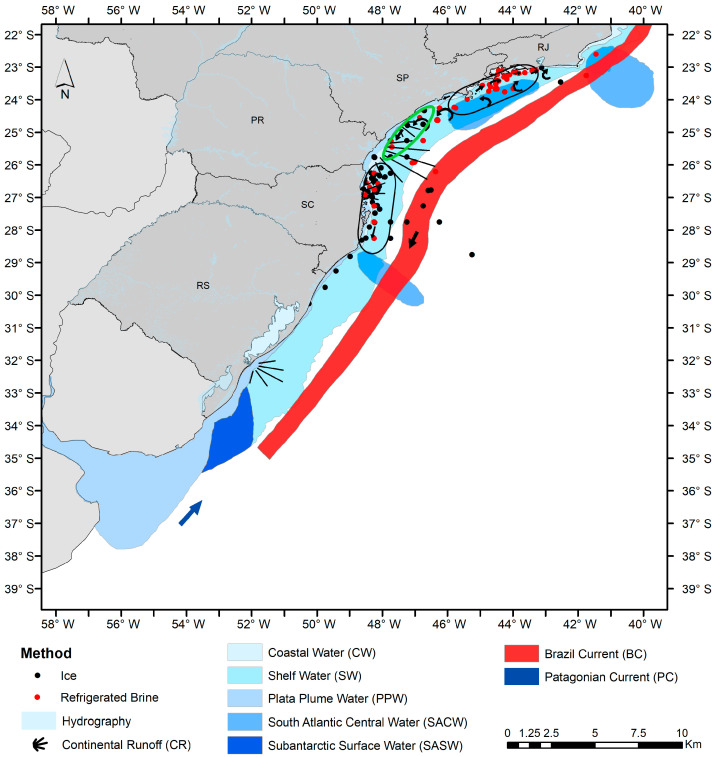
Spatial distribution of *Sardinella brasiliensis* along the southeast–south Brazilian coast and the major fishing grounds: Rio de Janeiro (RJ), São Paulo (SP), Paraná (PR), Santa Catarina (SC), and Rio Grande do Sul (RS). Black and green ellipses represent respectively the two spawning areas and the feeding ground for *S. brasiliensis* [21,46]. Colored areas represent the main oceanographic features in southeast–south Brazil: continental runoff, coastal water, shelf water, tropical water, Brazilian Current, South Atlantic Central Water, and Patagonian Current. Water mass fronts and the direction of the flow are represented by sets [47,48,49,50].

**Figure 2 biology-14-00013-f002:**
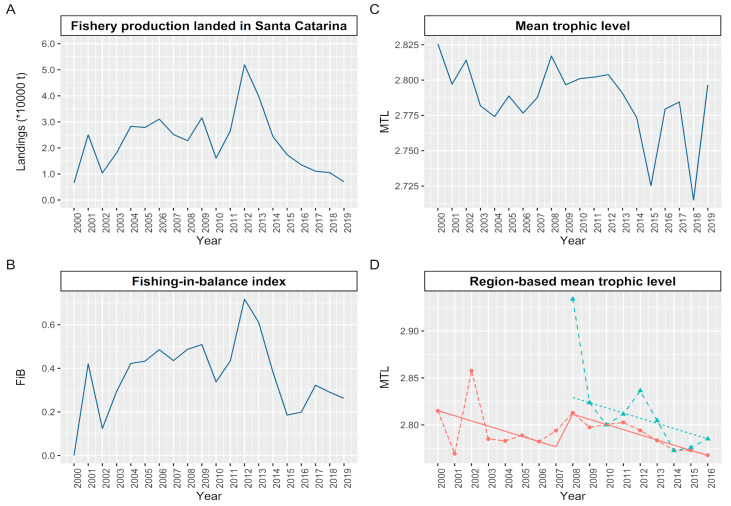
(**A**) Total fishery production of *S. brasiliensis* landed by the purse seine fleet in Santa Catarina between the years 2000 and 2019. (**B**) Fishing-in balance index that verifies the stability of the fishery in relation to the trophic level (FiB), (**C**) mean trophic level (MTL), and (**D**) region-based mean trophic level by fishing area. The dashed lines represent the trophic level per fishing area (RMTL) exploited by vessels with different conservation methods (VIC, ice fleet in red and VRB, refrigerated brine fleet in green) and the solid lines represent the RMTL adjusted by GAMLSS for each region.

**Figure 3 biology-14-00013-f003:**
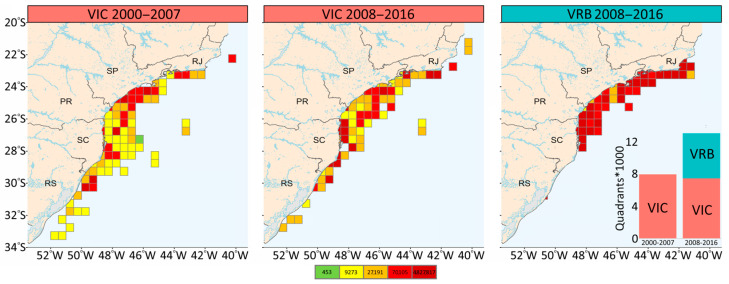
Spatial distribution of fishing trips of the purse seine fleet targeting the Brazilian sardine monitored in the main ports of landing in the state of Santa Catarina between 2000 and 2016. Quadrants represent catches of the ice fleet (VIC) and the refrigerated brine fleet (VRB) divided into quartiles.

**Figure 4 biology-14-00013-f004:**
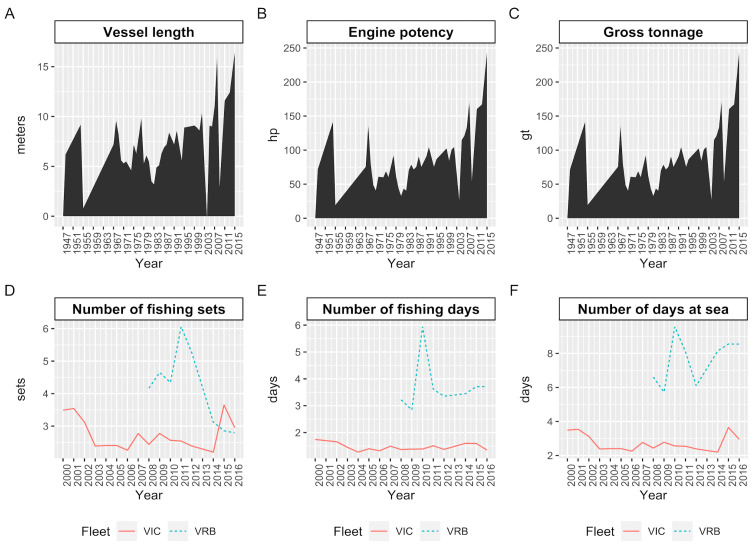
Evolution of physical characteristics of the purse seine fleet measured through information on (**A**) vessel length, (**B**) engine power, and (**C**) gross tonnage extracted from the general fishery register (1938–2015) and its influence on (**D**) number of fishing sets, (**E**) fishing days, and (**F**) days at sea between 2000 and 2016.

**Figure 5 biology-14-00013-f005:**
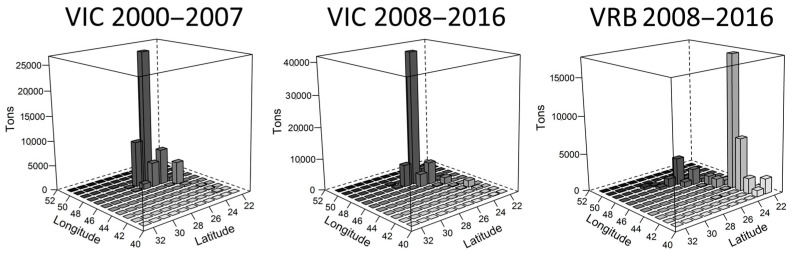
Abundance of monitored Brazilian sardines divided according to the expansion of the fishing area and fleet: VIC, original area 2000–2007; VIC, expanded area 2008–2016; VRB, expanded 2008–2016.

**Figure 6 biology-14-00013-f006:**
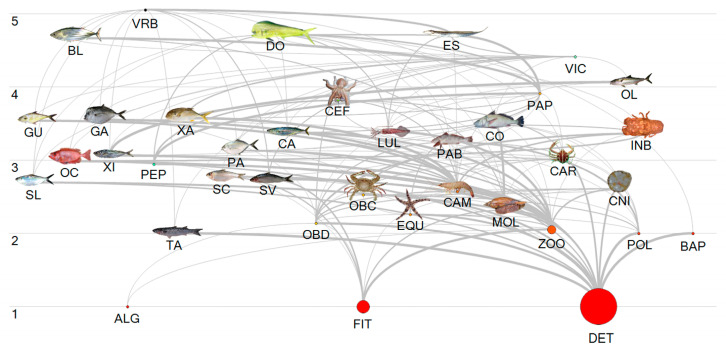
Diagram of energy flux (t km^−2^ year^−1^) from the main species captured by the purse seine fishery targeting *Sardinella brasiliensis* off southeast and south. The vertical axis indicates the trophic level of the represented species: SL, Atlantic thread herring; BL, skipjack tuna; DO, dolphinfish; ES, largehead hairtail; GU, castin leatherjacket; OL, yellowfin amberjack; XI, rough scad; OC, Atlantic bigeye; TA, lebranche mullet; GA, Atlantic moonfish; XA, crevalle jack; PA, Atlantic bumper; CA, Atlantic chub mackerel; SC, false herring; CO, whitemouth croaker; SV, Brazilian sardine, and the prey contained in their diet, PEP, pelagic fish; CEF, cephalopods; LUL, squids; PAB, benthic feeding fish; OBD, detritivores benthic organisms; OBC, carnivorous benthic organisms; PAP, pelagic feeding fish; CAR, crabs; EQU, echinoderms; INB, benthic invertebrates; CAM, shrimps; MOL, mollusks; ZOO, zooplankton; CNI, cnidarians; POL, polychaeta; BAP, bacterioplankton; ALG, algae; FIT, phytoplankton; DET, detritus.

**Figure 7 biology-14-00013-f007:**
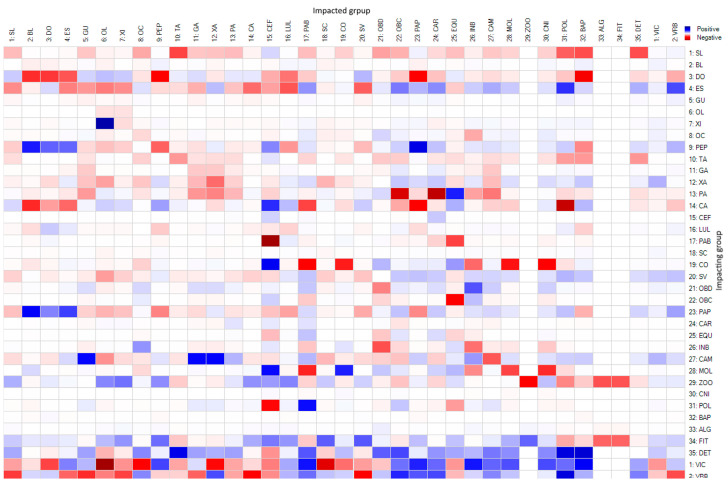
Direct and indirect impacts that increased biomass of impacting group species may exert on the Impacted group within the ecosystem exploited by the purse seine fishery in SSB. The colors in the squares represent the control effects ranging from positive bottom–up control effects in blue to negative top–down control effects in red. SL, Atlantic thread herring; BL, skipjack tuna; DO, dolphinfish; ES, largehead hairtail; GU, castin leatherjacket; OL, yellowfin amberjack; XI, rough scad; OC, Atlantic bigeye; TA, lebranche mullet; GA, Atlantic moonfish; XA, crevalle jack; PA, Atlantic bumper; CA, Atlantic chub mackerel; SC, false herring; CO, whitemouth croaker; SV, Brazilian sardine, and the prey contained in their diet, PEP, pelagic fish; CEF, cephalopods; LUL, squids; PAB, benthic feeding fish; OBD, detritivores benthic organisms; OBC, carnivorous benthic organisms; PAP, pelagic feeding fish; CAR, crabs; EQU, echinoderms; INB, benthic invertebrates; CAM, shrimps; MOL, mollusks; ZOO, zooplankton; CNI, cnidarians; POL, polychaeta; BAP, bacterioplankton; ALG, algae; FIT, phytoplankton; DET, detritus.

**Figure 8 biology-14-00013-f008:**
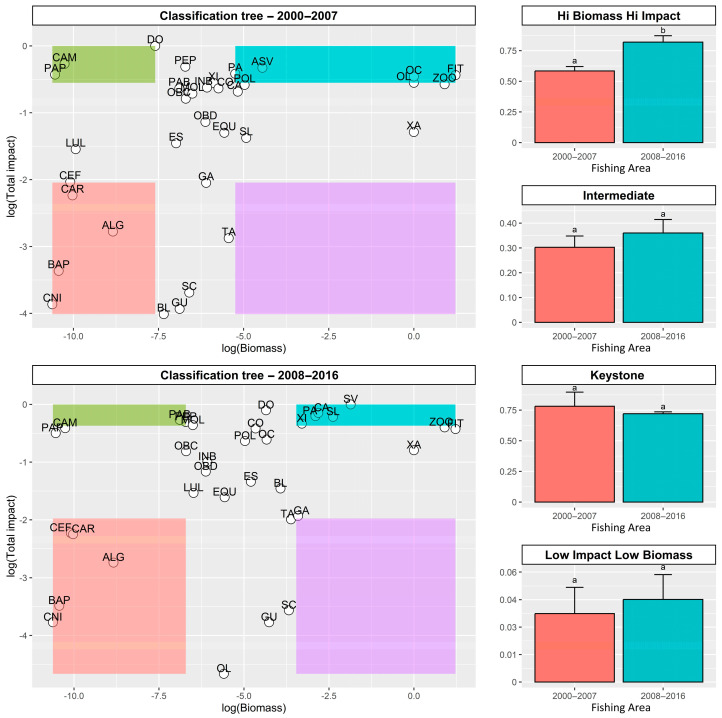
Application of the classification tree to the southeast and south Brazil food web for the original area (2000–2007), and for the expanded area (2008–2016). The scatterplot shows the log-transformed biomass (Bi) on the x-axis, and the log-transformed trophic impact (in squared values) on the y-axis. Each point is a functional group in the model, identified with a group name (indicated in the legend). Groups belonging to each category are displayed as the corresponding category box: high-impact–high-biomass, intermediate, keystone, and low-impact–low-biomass. Significant differences for each one of these categories were represented by different letters in the right-side bar charts, in which different letters represent significantly different means. SL, Atlantic thread herring; BL, skipjack tuna; DO, dolphinfish; ES, largehead hairtail; GU, castin leatherjacket; OL, yellowfin amberjack; XI, rough scad; OC, Atlantic bigeye; TA, lebranche mullet; GA, Atlantic moonfish; XA, crevalle jack; PA, Atlantic bumper; CA, Atlantic chub mackerel; SC, false herring; CO, whitemouth croaker; SV, Brazilian sardine, and the prey contained in their diet, PEP, pelagic fish; CEF, cephalopods; LUL, squids; PAB, benthic feeding fish; OBD, detritivores benthic organisms; OBC, carnivorous benthic organisms; PAP, pelagic feeding fish; CAR, crabs; EQU, echinoderms; INB, benthic invertebrates; CAM, shrimps; MOL, mollusks; ZOO, zooplankton; CNI, cnidarians; POL, polychaeta; BAP, bacterioplankton; ALG, algae; FIT, phytoplankton; DET, detritus.

**Figure 9 biology-14-00013-f009:**
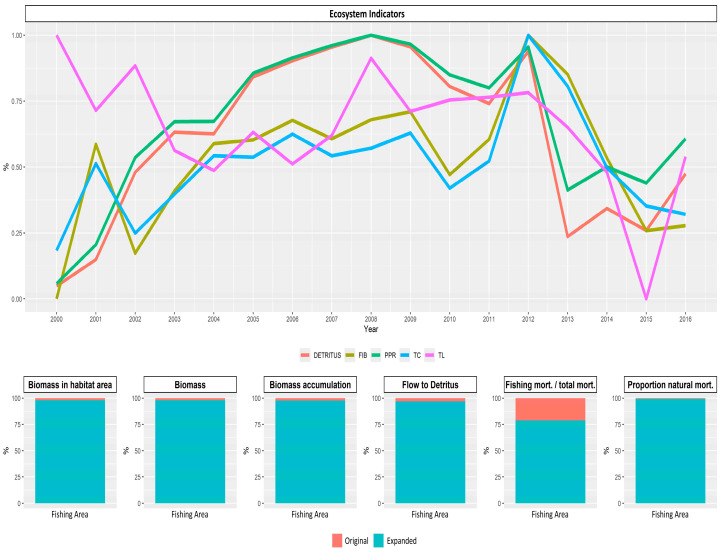
Temporal variability of the mean trophic level (MTL) alongside proportional variation of fishing in balance index (FiB), total catch (TC), primary production required to sustain catches (PPR), and flow to detritus (detritus) between 2000 and 2016 in southeast Brazil. Significant differences for significant EwE model variables were represented in the bottom-side bar charts.

**Table 1 biology-14-00013-t001:** Trophic level (TL) of the main species landed in the Brazilian sardine fishery in the southeast and south of Brazil. Values expressed were normalized for each species [56].

Class	Order	Family	Species	Common Name	TL
Cephalopoda	Teuthida	Loliginidae	*Doryteuthis plei*	Slender inshore squid	3.30
Elasmobranchii	Squaliformes	Squalidae	*Squalus* spp.	Dogfish	3.22
	Lamniformes	Odontaspididae	*Carcharias taurus*	Sand tiger shark	4.08
	Carchariniformes	Carcharinidae	*Carcharhinus brevipinna*	Spinner shark	3.87
			*Carcharhinus leucas*	Bull shark	5.00
			*Prionace glauca*	Blue shark	4.24
			*Rhizoprionodon porosus*	Carribean sharpnose shark	3.26
		Sphyrnidae	*Sphyrna lewini*	Scalloped hammerhead	3.82
	Rajiformes	Arhynchobatidae	*Atlantoraja castelnaui*	Spotback skate	4.28
			Not discriminated	Rays	4.28
Actinopterygii	Clupeiformes	Engraulidae	*Anchoviella lepidentostole*	Broadband anchovy	2.88
			*Engraulis anchoita*	Argentine anchovy	2.83
		Clupeidae	*Brevoortia spp*	Menhaden	3.10
			*Harengula clupeola*	False herring	3.05
			*Opisthonema oglinum*	Atlantic thread herring	2.61
			*Pellona harroweri*	American coastal pellona	3.33
			*Sardinella brasiliensis*	Brazilian sardine	2.77
	Siluriformes	Ariidae	Family Ariidae	Catfish	3.13
	Mugiliformes	Mugilidae	*Mugil liza*	Lebranche mullet	2.00
	Beloniformes	Belonidae	*Ablennes hians*	Flat needlefish	5.00
	Zeiformes	Zeidae	*Zenopsis conchifer*	Silvery John dory	3.10
	Scorpaeniformes	Scorpaenidae	*Helicolenus lahillei*	Rosefish	4.17
	Scorpaeniformes	Triglidae	*Prionotus punctatus*	Bluewing searobin	3.72
	Perciformes	Centropomidae	*Centropomus spp,*	Snook	3.19
		Serranidae	*Epinephelus marginatus*	Dusky grouper	3.94
		Priacanthidae	*Priacanthus arenatus*	Atlantic bigeye	2.99
		Pomatomidae	*Pomatomus saltatrix*	Bluefish	3.60
		Coryphaenidae	*Coryphaena hippurus*	Common Dolphinfish	4.44
		Carangidae	*Caranx crysos*	Blue runner	3.18
			*Caranx hippos*	Crevalle jack	3.24
			*Caranx latus*	Horse-eye jack	3.59
			*Chloroscombrus chrysurus*	Atlantic bumper	2.49
			*Oligoplites saliens*	Castin leatherjacket	3.28
			*Parona signata*	Parona leatherjacket	3.10
			*Selene setapinnis*	Atlantic moonfish	3.28
			*Seriola dumerili*	Greater amberjack	3.22
			*Trachurus lathami*	Rough scad	2.77
			*Seriola lalandi*	Yellowtail amberjack	3.39
Actinopterygii	Perciformes	Carangidae	*Trachinotus carolinus*	Florida pompano	2.75
		Lutjanidae	*Rhomboplites aurorubens*	Vermilion snapper	3.47
		Gerreidae	*Diapterus rhombeus*	Caitipa mojarra	2.82
			*Eucinostomus* spp.	Mojarra	3.04
		Haemulidae	*Conodon nobilis*	Bared grunt	3.29
			*Haemulon aurolineatum*	Tomtate grunt	2.82
		Sparidae	*Archosargus probatocephalus*	Sheepshead	2.88
			*Pagrus pagrus*	Red porgy	3.83
		Sciaenidae	*Cynoscion acoupa*	Acoupa weakfish	3.80
			*Cynoscion guatucupa*	Stripped weakfish	3.64
			*Cynoscion jamaicensis*	Jamaica weakfish	3.06
			*Cynoscion leiarchus*	Smooth weakfish	3.12
			*Cynoscion microlepidotus*	Smallscale weakfish	3.32
			*Cynoscion virescens*	Green weakfish	3.60
			*Larimus breviceps*	Shorthead drum	3.26
		Sciaenidae	*Macrodon atricauda*	Southern king weakfish	3.14
			*Mentichirrhus americanus*	Southern kingcroaker	3.23
			*Micropogonias furnieri*	Whitemouth croaker	3.11
			*Paralonchurus brasiliensis*	Banded croaker	3.14
			*Pogonias cromis*	Black drum	3.34
			*Stellifer rastrifer*	Rake stardrum	3.12
			*Umbrina canosai*	Argentine croaker	3.11
		Ephippidae	*Chaetodipterus faber*	Atlantic spadefish	2.96
		Trichiuridae	*Trichiurus lepturus*	Largehead hairtail	3.21
		Scombridae	*Auxis thazard*	Frigate tuna	3.50
			*Euthynnus alletteratus*	Little tuny	3.90
			*Katsuwonus pelamis*	Skipjack tuna	3.24
			Non discriminated	Tuna fish	3.19
			*Scomber colias*	Atlantic chub mackerel	2.99
			*Scomberomorus brasiliensis*	Serra Spanish mackerel	3.04
			*Thunnus albacares*	Yellowfin tuna	4.27
		Xiphiidae	*Xiphias gladius*	Swordfish	3.93
		Istiophoridae	*Istiophorus albicans*	Atlantic sailfish	4.19
		Stromateidae	*Peprilus paru*	American harvestfish	2.77
	Tetraodontiformes	Balistidae	*Balistes capriscus*	Grey triggerfish	3.40

**Table 2 biology-14-00013-t002:** Basic input and output parameters for the Ecopath model of the southeast and southern Brazil ecosystem. TL, trophic level, B, biomass (t km^−2^), PB, production/biomass (year^−1^), QB, consumption/biomass (year^−1^), EE, ecotrophic efficiency, PQ production/consumption, OI, omnivory Index, Total catches for the ice fleet (VIC) and the refrigerated brine fleet (VRB) in the original and expanded fishing area (t km^−2^ year^−1^).

Code	Common Name	Fishing Area	TL	B	PB	QB	EE	PQ	OI	VIC	VRB
1. SL	Atlantic thread herring	Original	2.69	1.18 × 10^−02^	0.580	1.619	0.868	3.58 × 10^−01^	0.249	3.74 × 10^−03^	-
2. BL	Skipjack tuna		4.67	4.52 × 10^−05^	0.340	0.749	0.772	4.53 × 10^−01^	0.324	3.19 × 10^−05^	-
3. DO	Dolphinfish		4.72	2.52 × 10^−05^	0.180	1.212	0.772	1.48 × 10^−01^	0.480	2.50 × 10^−05^	-
4. ES	Largehead hairtail		4.71	1.04 × 10^−04^	0.360	0.661	0.772	5.44 × 10^−01^	0.177	8.27 × 10^−06^	-
5. GU	Castin leatherjacket		3.54	1.32 × 10^−04^	0.480	0.661	0.868	7.26 × 10^−01^	0.079	4.61 × 10^−05^	-
6. OL	Yellowtail amberjack		4.07	1.92 × 10^−06^	0.180	1.212	0.772	1.48 × 10^−01^	0.001	0	-
7. XI	Rough scad		3.06	8.93 × 10^−05^	0.430	1.092	0.868	3.93 × 10^−01^	0.011	3.74 × 10^−04^	-
8. OC	Atlantic bigeye		2.99	4.62 × 10^−05^	0.570	5.300	0.648	1.07 × 10^−01^	1.000	0	-
9. PEP	Pelagic fish		2.95	1.93 × 10^−07^	2.500	25.000	0.742	1.00 × 10^−01^	0.100	0	-
10. TA	Lebranche mullet		2.00	3.64 × 10^−03^	0.320	1.296	0.648	2.46 × 10^−01^	0	5.90 × 10^−04^	-
11. GA	Atlantic moonfish		3.55	7.93 × 10^−04^	0.750	0.987	0.868	7.59 × 10^−01^	0.067	1.36 × 10^−04^	-
12. XA	Crevalle jack		3.58	1.62 × 10^−07^	0.570	5.300	0.868	1.07 × 10^−01^	0.0	3.20 × 10^−04^	-
13. PA	Atlantic bumper		3.13	5.66 × 10^−03^	0.400	0.943	0.868	4.24 × 10^−01^	0.462	8.65 × 10^−04^	-
14. CA	Atlantic chub mackerel		3.38	6.73 × 10^−03^	0.480	0.950	0.868	5.05 × 10^−01^	0.492	4.51 × 10^−04^	-
15. CEF	Cephalopods		3.82	8.10 × 10^−11^	3.000	10.000	0.858	3.00 × 10^−01^	0.282	0	-
16. LUL	Squids		3.39	1.16 × 10^−07^	0.680	5.155	0.858	1.31 × 10^−01^	0.476	1.16 × 10^−07^	-
17. PAB	Benthic feeding fish		3.25	1.28 × 10^−07^	0.960	7.600	0.648	1.26 × 10^−01^	0.144	0	-
18. SC	False herring		2.77	2.09 × 10^−04^	0.930	1.500	0.868	6.20 × 10^−01^	0.288	0	-
19. CO	Whitemouth croaker		3.48	1.82 × 10^−03^	0.340	0.801	0.800	4.24 × 10^−01^	0.149	6.41 × 10^−04^	-
20. SV	Brazilian sardine		2.74	2.53 × 10^−04^	0.810	1.714	0.868	4.72 × 10^−01^	0.231	2.57 × 10^−04^	-
21. OBD	Detritivorous benthic organisms		2.14	7.53 × 10^−07^	7.860	13.450	0.501	5.84 × 10^−01^	0.130	0	-
22. OBC	Carnivorous benthic organisms		2.53	1.99 × 10^−07^	6.630	12.220	0.711	5.42 × 10^−01^	0.354	0	-
23. PAP	Pelagic feeding fish		3.91	2.90 × 10^−11^	1.300	2.800	0.868	4.64 × 10^−01^	0.018	0	-
24. CAR	Crabs		3.01	9.40 × 10^−11^	4.420	25.800	0.729	1.71 × 10^−01^	0.905	0	-
25. EQU	Echinoderms		2.26	2.70 × 10^−06^	1.580	2.860	0.137	5.52 × 10^−01^	0.214	0	-
26. INB	Benthic invertebrates		3.37	8.38 × 10^−07^	2.680	3.894	0.950	6.88 × 10^−01^	0.044	0	-
27. CAM	Shrimps		2.58	5.50 × 10^−11^	3.930	19.130	0.622	2.05 × 10^−01^	0.297	0	-
28. MOL	Mollusks		2.32	3.12 × 10^−07^	5.290	8.210	0.950	6.44 × 10^−01^	0.233	0	-
29. ZOO	Zooplankton		2.05	8.00 × 10^0^	40.000	324.600	0.429	1.23 × 10^−01^	0.053	0	-
30. CNI	Cnidaria		2.61	2.40 × 10^−11^	1.000	2.000	0.551	5.00 × 10^−01^	0.311	0	-
31. POL	Polychaeta		2.00	1.07 × 10^−05^	6.320	11.190	0.205	5.64 × 10^−01^	0	0	-
32. BAP	Bacterioplankton		2.00	3.70 × 10^−11^	250.000	500.000	0.689	5.00 × 10^−01^	0	0	-
33. ALG	Algae		1.00	1.44 × 10^−09^	41.660	−	0.950	−	0	0	-
34. FIT	Fitoplankton		1.00	1.67 × 10^+01^	100,000.000	−	0.965	−	0	0	-
35. DET	Detritus		1.00	1.50 × 10^+02^	−	−	0.000		0.013	0	-
1. SL	Atlantic thread herring	Expanded	2.69	4.16 × 10^−03^	0.580	1.619	0.868	3.58 × 10^−01^	0.249	4.09 × 10^−03^	6.88 × 10^−04^
2. BL	Skipjack tuna		4.67	1.18 × 10^−04^	0.340	0.749	0.772	4.53 × 10^−01^	0.324	1.02 × 10^−04^	5.49 × 10^−06^
3. DO	Dolphinfish		4.72	4.44 × 10^−05^	0.180	1.212	0.772	1.48 × 10^−01^	0.480	2.92 × 10^−05^	1.69 × 10^−06^
4. ES	Largehead hairtail		4.71	1.59 × 10^−05^	0.360	0.661	0.772	5.44 × 10^−01^	0.177	8.69 × 10^−06^	1.25 × 10^−05^
5. GU	Castin leatherjacket		3.54	5.50 × 10^−05^	0.480	0.661	0.868	7.26 × 10^−01^	0.079	6.29 × 10^−06^	1.08 × 10^−04^
6. OL	Yellowtail amberjack		4.07	2.56 × 10^−06^	0.180	1.212	0.772	1.48 × 10^−01^	0.001	2.56 × 10^−06^	0
7. XI	Rough scad		3.06	5.04 × 10^−04^	0.430	1.092	0.868	3.93 × 10^−01^	0.011	3.74 × 10^−04^	8.93 × 10^−05^
8. OC	Atlantic bigeye		2.99	4.62 × 10^−05^	0.570	5.300	0.648	1.07 × 10^−01^	1.000	4.60 × 10^−05^	0
9. PEP	Pelagic fish		2.95	1.93 × 10^−07^	2.500	25.000	0.742	1.00 × 10^−01^	0.100	0	0
10. TA	Lebranche mullet		2.00	2.40 × 10^−04^	0.320	1.296	0.648	2.46 × 10^−01^	0	1.42 × 10^−04^	3.50 × 10^−05^
11. GA	Atlantic moonfish		3.55	3.89 × 10^−04^	0.750	0.987	0.868	7.59 × 10^−01^	0.067	3.72 × 10^−04^	2.21 × 10^−04^
12. XA	Crevalle jack		3.58	6.13 × 10^−04^	0.570	5.300	0.868	1.07 × 10^−01^	0	0	0
13. PA	Atlantic bumper		3.13	1.26 × 10^−03^	0.400	0.943	0.868	4.24 × 10^−01^	0.462	1.35 × 10^−03^	1.55 × 10^−04^
14. CA	Atlantic chub mackerel		3.38	1.53 × 10^−03^	0.480	0.950	0.868	5.05 × 10^−01^	0.492	7.66 × 10^−04^	9.00 × 10^−04^
15. CEF	Cephalopods		3.82	8.10 × 10^−11^	3.000	10.000	0.858	3.00 × 10^−01^	0.282	0	0
16. LUL	Squids		3.39	3.29 × 10^−07^	0.680	5.155	0.858	1.31 × 10^−01^	0.476	3.29 × 10^−07^	0
17. PAB	Benthic feeding fish		3.25	1.28 × 10^−07^	0.960	7.600	0.648	1.26 × 10^−01^	0.144	0	0
18. SC	False herring		2.77	2.09 × 10^−04^	0.930	1.500	0.868	6.20 × 10^−01^	0.288	2.09 × 10^−04^	0
19. CO	Whitemouth croaker		3.48	2.19 × 10^−05^	0.340	0.801	0.800	4.24 × 10^−01^	0.149	9.35 × 10^−06^	1.00 × 10^−05^
20. SV	Brazilian sardine		2.74	1.37 × 10^−02^	0.810	1.714	0.868	4.72 × 10^−01^	0.231	1.24 × 10^−02^	5.02 × 10^−03^
21. OBD	Detritivorous benthic organisms		2.14	7.53 × 10^−07^	7.860	13.450	0.501	5.84 × 10^−01^	0.130	0	0
22. OBC	Carnivorous benthic organisms		2.53	1.99 × 10^−07^	6.630	12.220	0.711	5.42 × 10^−01^	0.354	0	0
23. PAP	Pelagic feeding fish		3.91	2.90 × 10^−11^	1.300	2.800	0.868	4.64 × 10^−01^	0.018	0	0
24. CAR	Crabs		3.01	9.40 × 10^−11^	4.420	25.800	0.729	1.71 × 10^−01^	0.905	0	0
25. EQU	Echinoderms		2.26	2.70 × 10^−06^	1.580	2.860	0.137	5.52 × 10^−01^	0.214	0	0
26. INB	Benthic invertebrates		3.37	8.38 × 10^−07^	2.680	3.894	0.950	6.88 × 10^−01^	0.044	0	0
27. CAM	Shrimps		2.58	5.50 × 10^−11^	3.930	19.130	0.622	2.05 × 10^−01^	0.297	0	0
28. MOL	Mollusks		2.32	3.12 × 10^−07^	5.290	8.210	0.950	6.44 × 10^−01^	0.233	0	0
29. ZOO	Zooplankton		2.05	8.00 × 10^0^	40.000	324.600	0.429	1.23 × 10^−01^	0.053	0	0
30. CNI	Cnidaria		2.61	2.40 × 10^−11^	1.000	2.000	0.551	5.00 × 10^−01^	0.311	0	0
31. POL	Polychaeta		2.00	1.07 × 10^−05^	6.320	11.190	0.205	5.64 × 10^−01^	0	0	0
32. BAP	Bacterioplankton		2.00	3.70 × 10^−11^	250.000	500.000	0.689	5.00 × 10^−01^	0	0	0
33. ALG	Algae		1.00	1.44 × 10^−09^	41.660	−	0.950	−	0	0	0
34. FIT	Fitoplankton		1.00	1.67 × 10^+01^	100,000.000	−	0.965	−	0	0	0
35. DET	Detritus		1.00	1.50 × 10^+02^	67.500	−	−	5.24 × 10^−08^	0.013	0	0

## Data Availability

Data will be available under a reasonable request.

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
