# Peer review of "The History of the Brazilian Sardine (Sardinella brasiliensis) Between Two Fishery Collapses: An Ecosystem Modeling Approach to Study Its Life Cycle"

_biology, 2024, doi:10.3390/biology14010013_

Round 1
Reviewer 1 Report
Comments and Suggestions for Authors
The manuscript "The history of the Brazilian sardine (Sardinella brasiliensis) between two fishery collapses: an ecosystem modelling approach to study its life cycle," is very interesting and well-designed. The paper can be accepted after major revision.
1. Too many references: For a review paper, more than 70-80 references are too many. Also, many old references were cited; revise it
2. Several paces words are merged (e.g., line 26).
3. The degree sign is not correct; check throughout the manuscript (line 120, 122, 123, 126, etc.).
4. While calculating the trophic level of the catches on a spatial and temporal scale, why were only data between 2000-2019 used, but not up to 2024?
5. Similarly, for other data, e.g., S. brasiliensis catch, only data between 2008 and 2016 were used.
Author Response
Answers of the authors the reviewer´s comments
# Reviewer 1
- Too many references: For a review paper, more than 70-80 references are too many. Also, many old references were cited; revise it
Suggestions accepted. See the revised MS. References were reduce from 120 to 85.
- Several paces words are merged (e.g., line 26).
Suggestions accepted. See the revised MS, line 36.
- The degree sign is not correct; check throughout the manuscript (line 57, 58, 120, 122, 123, 126, etc.).
Suggestions accepted. See the revised MS line 66, 67, 69, 70, 149, 160, 161, 163, 343, 440, 441, 462, 464, 465, 466, and 680.
- While calculating the trophic level of the catches on a spatial and temporal scale, why were only data between 2000-2019 used, but not up to 2024?
Because the aim was to study the period between collapses. After the collapse of the Brazilian sardine, the fleet moved on to other species, especially the Atlantic thread herring, so we would no longer be studying the behavior of the Brazilian sardine fishery. See the revised MS line 183-199.
- Similarly, for other data, e.g., S. brasiliensis catch, only data between 2008 and 2016 were used.
No. Catch information were included from 2000 to 2019. For the refrigerated brine fleet, where the vessels were built and started operating after 2008, yes, the data was obtained from 2008 onwards.
Reviewer 2 Report
Comments and Suggestions for Authors
A brief summary:
This MS by Schroeder et al. evaluated the trophic relationships of the main species exploited by this fishery and the importance of its biomass for the southeast-south Brazil marine ecosystem (22ºS-34oS). Data was analyzed using a mass balance model (ECOPATH) between the two fishery collapses: 2000 and 2017. From 2000 onwards, the sardine fishery adopted a multi-species character. The mean trophic level of the catches (MTL) showed a decreasing trend until 2008, when more modern vessels with greater autonomy have entered the fishery, and expanded the traditional fishing area to exploit northern fishing grounds. The MTL in the expanded fishing area suddenly increase, and has characterized by high biomass of the Brazilian sardine and other species with high biomass and high eco-trophic impact, felling again to the lowest level in2016. The model evidenced high estimates for fishing mortality, natural mortality, and flow to detritus between 2008 and 2016, when sardine fishing collapsed. During this period, a sharp drop in the primary production required to sustain the catches from 2012 onwards accompanied of a significant fall in the biomass accumulation rate.
Overall, I thought this was a well-executed study in a system with limited previous knowledge of this level on this specific topic. I appreciated their multi-faceted approach and the time course involved in this work and think it add significant merit to their work. I do think that their overall conclusions were related to results. Overall, I think this is good work, but should undergo some revision before acceptance.
1) General concept comments.
I think the author should seek help with writing in English, I hope round of editing with English native language would further improve the quality of the writing. The English grammar should be checked again.
I also have a couple suggestions.
2) Specific comments
Line 365-370 the author explained Fig. 2A, 2B and 2C, but they explained Fig. 2D until line 416-417. Why did not the author explain Fig. 2A, 2B 2C and 2D together? Is it necessary to insert so many other contents in the between 2A, 2B 2C and 2D?
Line 38-54, Incorrect citation format for references.
Line 488 In Fig 8, The author should indicate which picture is A? which picture is B? which picture is C to make it clearer. Because they marked Fig 8A, B, C in the text (Line 477-485)
As an article type, there are as much as 120 References, even more than a review.
I suggest author delete some unimportant or too old References.
Comments on the Quality of English Language
The English could be improved to more clearly express the research.
Author Response
Answers of the authors the reviewer´s comments
# Reviewer 2
A brief summary: This MS by Schroeder et al. evaluated the trophic relationships of the main species exploited by this fishery and the importance of its biomass for the southeast-south Brazil marine ecosystem (22ºS-34oS). Data was analyzed using a mass balance model (ECOPATH) between the two fishery collapses: 2000 and 2017. From 2000 onwards, the sardine fishery adopted a multi-species character. The mean trophic level of the catches (MTL) showed a decreasing trend until 2008, when more modern vessels with greater autonomy have entered the fishery, and expanded the traditional fishing area to exploit northern fishing grounds. The MTL in the expanded fishing area suddenly increase, and has characterized by high biomass of the Brazilian sardine and other species with high biomass and high eco-trophic impact, felling again to the lowest level in2016. The model evidenced high estimates for fishing mortality, natural mortality, and flow to detritus between 2008 and 2016, when sardine fishing collapsed. During this period, a sharp drop in the primary production required to sustain the catches from 2012 onwards accompanied of a significant fall in the biomass accumulation rate. Overall, I thought this was a well-executed study in a system with limited previous knowledge of this level on this specific topic. I appreciated their multi-faceted approach and the time course involved in this work and think it add significant merit to their work. I do think that their overall conclusions were related to results. Overall, I think this is good work, but should undergo some revision before acceptance.
Thank you for you positive feed-back.
1) General concept comments.
I think the author should seek help with writing in English, I hope round of editing with English native language would further improve the quality of the writing. The English grammar should be checked again.
- Suggestion accepted. We revised the MS and enhanced the quality of English.;
I also have a couple suggestions.
2) Specific comments
Line 365-370 the author explained Fig. 2A, 2B and 2C, but they explained Fig. 2D until line 416-417. Why did not the author explain Fig. 2A, 2B 2C and 2D together? Is it necessary to insert so many other contents in the between 2A, 2B 2C and 2D?
- Suggestion accepted. See revised MS line 420-426.
Line 38-54, Incorrect citation format for references.
- Suggestion accepted. See the revised MS line 48, 52, 54,56, 73, 118-119, 159, 696.
Line 488 In Fig 8, The author should indicate which picture is A? which picture is B? which picture is C to make it clearer. Because they marked Fig 8A, B, C in the text (Line 477-485)
- Suggestion accepted. The text was revised. See revised MS line 541.
As an article type, there are as much as 120 References, even more than a review. I suggest author delete some unimportant or too old References.
- Suggestions accepted. See revised MS.
Reviewer 3 Report
Comments and Suggestions for Authors
Review for the paper “The history of the Brazilian sardine (Sardinella brasiliensis) between two fishery collapses: an ecosystem modelling approach to study its life cycle” by Rafael Schroeder and co-authors submitted to “Biology”.
The authors of this research paper conducted an analysis of the inter-annual fluctuations in the abundance of Brazilian sardine (Sardinella brasiliensis) and their implications for the purse seine fishery dynamics along the southeast-south coast of Brazil. Using a mass balance model ECOPATH, the study spanned two periods of fishery collapse around the turn of the millennium and in 2017. The results of this study may have important implications for understanding the sustainability of the fishery. The period leading up to the sardine collapse was characterized by a stark decline in primary production necessary to support the catches, as well as a decrease in biomass accumulation rates. This environmental shift occurred in conjunction with rising sea temperatures, which were identified as a potential limiting factor affecting primary productivity.
Furthermore, the study suggests that increasing water temperatures likely contributed to natural mortality rates and the flow of biomass into detritus. The findings show how intensifying fishing pressures may have exacerbated the challenges faced by the sardine population, leading to its collapse.
The paper is well-written and well-illustrated. The authors have employed standard statistical methods to compare data and substantiate their conclusions. I have only a few suggestions to further enhance the paper.
Recommendations.
The authors should provide a simple summary of their main findings, as this section is mandatory for this journal.
Introduction.
L 44-49. The authors should clarify how the feeding habits and trophic positions of sardines influence their role in marine food chains, including specific examples of predator-prey interactions that might illustrate this. What factors contribute to the spawning success of Brazilian sardines in the identified areas?
L 54-59. The authors should provide more information on the environmental conditions that led to the historical collapse of sardine stocks. What role do climate change and oceanic conditions play in these fluctuations?
L 74. What evidence supports the idea that large, long-lived fish decline more rapidly than smaller, shorter-lived species in multispecies fisheries?
Materials and Methods.
L 147-152. The authors should describe in more detail the methods used to collect biological and fishery information during sampling of the industrial purse seine fleet. They should clarify how these data were validated for accuracy and reliability.
Section 2.2.2. The authors should mention how the equations and models used to determine trophic levels and mean trophic levels account for temporal variation in species availability.
L 221. What interviews did the authors refer to here?
L 257. The authors should describe in more detail how the normalized diet compositions for the species caught by the purse seine fleet were compiled. What methods did the authors use to ensure the accuracy of these dietary assessments?
L 360. What metrics did the authors use for the PERMANOVA analysis?
Results.
L 369-370, 391-397, 404-408, 510-513. Did the authors perform a statistical analysis to confirm the temporal trends described in the text? If so, what p-values did they obtain?
Section 3.3. The authors should use full species names instead of abbreviations for better understanding.
Discussion.
L 521-530. The authors should clarify how the different biological population units and their spatial/temporal distribution patterns influence the MTL observed in S. brasiliensis catches.
L 555-566. The text mentions an association between declines in MTL and declines in catches in heavily exploited areas such as the Mediterranean. It would be useful to discuss what lessons can be learned from this region and applied to the management of the fisheries of southeastern and southern Brazil in terms of sustainable practices. How the "fishing through the food web" effect can be addressed within the current fishery management frameworks to ensure balance within the pelagic ecosystems of southeast-south Brazil.
L 567-585. Given that the expansion of fishing areas may have contributed to the collapse of S. brasiliensis stocks, it would be useful to discuss the strategies that could be used to monitor and manage fishing pressure in these newly exploited regions.
L 617-637. The authors should mention the implications of the poleward distribution shift of fish fauna for the future of commercial fisheries in southeastern Brazil.
Finally, it would be useful to outline an action plan that includes adaptive management frameworks for fisheries that take into account changing ecological conditions for sustainable practices in the industry.
Author Response
Answers of the authors the reviewer´s comments
# Reviewer 3
Review for the paper “The history of the Brazilian sardine (Sardinella brasiliensis) between two fishery collapses: an ecosystem modelling approach to study its life cycle” by Rafael Schroeder and co-authors submitted to “Biology”. The authors of this research paper conducted an analysis of the inter-annual fluctuations in the abundance of Brazilian sardine (Sardinella brasiliensis) and their implications for the purse seine fishery dynamics along the southeast-south coast of Brazil. Using a mass balance model ECOPATH, the study spanned two periods of fishery collapse around the turn of the millennium and in 2017. The results of this study may have important implications for understanding the sustainability of the fishery. The period leading up to the sardine collapse was characterized by a stark decline in primary production necessary to support the catches, as well as a decrease in biomass accumulation rates. This environmental shift occurred in conjunction with rising sea temperatures, which were identified as a potential limiting factor affecting primary productivity. Furthermore, the study suggests that increasing water temperatures likely contributed to natural mortality rates and the flow of biomass into detritus. The findings show how intensifying fishing pressures may have exacerbated the challenges faced by the sardine population, leading to its collapse. The paper is well-written and well-illustrated. The authors have employed standard statistical methods to compare data and substantiate their conclusions. I have only a few suggestions to further enhance the paper.
Thank you for you positive feed-back.
Recommendations.
The authors should provide a simple summary of their main findings, as this section is mandatory for this journal.
- Suggestion accepted. See revised MS line 17-24.
Introduction.
L 44-49. The authors should clarify how the feeding habits and trophic positions of sardines influence their role in marine food chains, including specific examples of predator-prey interactions that might illustrate this. What factors contribute to the spawning success of Brazilian sardines in the identified areas?
- Suggestion accepted. See revised MS line 56-64.
L 54-59. The authors should provide more information on the environmental conditions that led to the historical collapse of sardine stocks. What role do climate change and oceanic conditions play in these fluctuations?
- Suggestion accepted. See revised MS line 76-83.
L 74. What evidence supports the idea that large, long-lived fish decline more rapidly than smaller, shorter-lived species in multispecies fisheries?
- Suggestion accepted. See revised MS line 100-104.
Materials and Methods.
L 147-152. The authors should describe in more detail the methods used to collect biological and fishery information during sampling of the industrial purse seine fleet. They should clarify how these data were validated for accuracy and reliability.
- Suggestion accepted. See revised MS line 183-199.
Section 2.2.2. The authors should mention how the equations and models used to determine trophic levels and mean trophic levels account for temporal variation in species availability.
- Suggestion accepted. See revised MS line 242.
L 221. What interviews did the authors refer to here?
- Suggestion accepted. The text was revised. See revised MS line 193.
L 257. The authors should describe in more detail how the normalized diet compositions for the species caught by the purse seine fleet were compiled. What methods did the authors use to ensure the accuracy of these dietary assessments?
- Suggestion accepted. See revised MS line 301-305.
L 360. What metrics did the authors use for the PERMANOVA analysis?
- Suggestion accepted. The text was revised. See revised MS line 409-411.
Results.
L 369-370, 391-397, 404-408, 510-513. Did the authors perform a statistical analysis to confirm the temporal trends described in the text? If so, what p-values did they obtain?
- No, these figures are mainly descriptive. However, technological differences between both fleets were analyzed by Schroeder, R.; Correia, A.T.; Medeiros, S.D.; Pessatti, M.L.; Schwingel, P.R. Spatiotemporal variability of the catch composition and discards estimates of the different methods of onboard preservation for the Brazilian sardine fishery in the southwest Atlantic Ocean. Thalassas 2022, 38, 573-597. https://doi.org/10.1007/s41208- 022-00398-5.
Section 3.3. The authors should use full species names instead of abbreviations for better understanding.
- Suggestion accepted. See revised MS line 524-533.
Discussion.
L 521-530. The authors should clarify how the different biological population units and their spatial/temporal distribution patterns influence the MTL observed in S. brasiliensis catches.
- Suggestion accepted. See revised MS line 577-582.
L 555-566. The text mentions an association between declines in MTL and declines in catches in heavily exploited areas such as the Mediterranean. It would be useful to discuss what lessons can be learned from this region and applied to the management of the fisheries of southeastern and southern Brazil in terms of sustainable practices. How the "fishing through the food web" effect can be addressed within the current fishery management frameworks to ensure balance within the pelagic ecosystems of southeast-south Brazil.
- Suggestion accepted. See revised MS line 620-624.
L 567-585. Given that the expansion of fishing areas may have contributed to the collapse of S. brasiliensis stocks, it would be useful to discuss the strategies that could be used to monitor and manage fishing pressure in these newly exploited regions.
- Suggestion accepted. See revised MS line 643-647.
L 617-637. The authors should mention the implications of the poleward distribution shift of fish fauna for the future of commercial fisheries in southeastern Brazil.
- Suggestion accepted. See revised MS line 700-705.
Finally, it would be useful to outline an action plan that includes adaptive management frameworks for fisheries that take into account changing ecological conditions for sustainable practices in the industry.
- Suggestion accepted. See revised MS line 721-736.
Round 2
Reviewer 1 Report
Comments and Suggestions for Authors
Paper can be accepted for publication